# NON-INTRUSIVE ADAPTATION: INPUT-CENTRIC PARAMETER-EFFICIENT FINE-TUNING FOR VERSATILE MULTIMODAL MODELING

## ABSTRACT

Large language models (LLMs) and vision language models (VLMs) demonstrate excellent performance on a wide range of tasks by scaling up parameter counts from $O(10^9)$ to $O(10^{12})$ levels and further beyond. These large scales make it impossible to adapt and deploy fully specialized models given a task of interest. Parameter-efficient fine-tuning (PEFT) emerges as a promising direction to tackle the adaptation and serving challenges for such large models. We categorize PEFT techniques into two types: intrusive and non-intrusive. Intrusive PEFT techniques directly change a model's internal architecture. Though more flexible, they introduce significant complexities for training and serving. Non-intrusive PEFT techniques leave the internal architecture unchanged and only adapt model-external parameters, such as embeddings for input. In this work, we describe `AdaLink` as a non-intrusive PEFT technique that achieves competitive performance compared to SoTA intrusive PEFT (LoRA) and full model fine-tuning (FT) on various tasks. We evaluate using both text-only and multimodal tasks, with experiments that account for both parameter-count scaling and training regime (with and without instruction tuning).

## 1 INTRODUCTION

While large language models (LLMs) (Vaswani et al., 2017; Raffel et al., 2020; Brown et al., 2020; Chowdhery et al., 2022; OpenAI, 2023; Anil et al., 2023) and vision-language models (VLMs) (Alayrac et al., 2022; Li et al., 2023; Wang et al., 2022a; Chen et al., 2023b) have recently demonstrated remarkable capabilities across a variety of tasks, several challenges persist. Due to the prohibitive engineering cost and inefficiencies involved in maintaining separate models for different tasks, it's still an open question how to adapt these models for different specialized use cases to incorporate the latest information. Therefore, there is a trend towards parameter-efficient fine-tuning (PEFT) as a promising solution to these challenges, offering a trade-off between adaptability and efficiency. PEFT techniques, such as adapters (Houlsby et al., 2019; Pfeiffer et al., 2020, 2021), LoRA (Hu et al., 2021), and prompt tuning (Lester et al., 2021b; Liu et al., 2021), introduce only a small percentage of additional parameters for fine-tuning while leaving the bulk of the LLM's parameters unchanged. Within this framework, we differentiate between intrusive and non-intrusive PEFT methods based on the degree to which they interact with or alter the LLM's core architecture, like the transformer blocks.

Intrusive adaptation methods, including LoRA (Hu et al., 2021), Adapter (Pfeiffer et al., 2021; Beck et al., 2021), prefix-tuning (Li & Liang, 2021) and their combinational methods (Chen et al., 2023a; Mao et al., 2021), make direct changes to the model architecture or the internal parameters flexibly, modifying the existing layers and adding new layers. While offering strong expressive power by flexibility and potentially reducing the performance gap akin to full model fine-tuning, they introduce significant complexities in architecture design spaces and the serving infrastructures. Moreover, these core architectural changes often lead to compatibility issues and complicate the engineering required for the deployment of a single LLM equipped with multiple adaptation components. Such intricacies also heighten the possibility of unintended behaviors, for instance, potentially loading incorrect adaptation weights for different tasks or layers, thereby making extensive validation and testing all the more imperative for ensuring model reliability.

In contrast, non-intrusive adaptation strategies like prompt-tuning (Lester et al., 2021b) aim to adjust a model's behavior with minimal changes to the internal architecture or parameters that are often achieved by modifying the input to the core architecture. They typically allow users to make granular changes at the input level for each example in the same batch. As a result, the model remains flexible and adaptable to different customization needs. However, non-intrusive Parameter Efficient Fine-Tuning (PEFT) methods such as prompt-tuning have encountered optimization challenges Razdaibiedina et al. (2023). They are often less effective in adapting models for complex tasks, such as multi-tasking (Wang et al., 2022c), and are still in the exploratory phase for multimodal settings, particularly in preserving the position of vision tokens when processing visual input. Toward these challenges, we introduce a novel approach called AdaLink that introduces an adaptation module situated between the embedding and main transformer blocks of LLMs form as a link, retaining the non-intrusive benefits and alleviating the optimization difficulties.

Recent work (Wei et al., 2021; Sanh et al., 2021; Mishra et al., 2022; Touvron et al., 2023) has demonstrated the ability of large language models (LLMs) to acquire a variety of skills and generalize well to unseen tasks through instruction tuning. In this paper, we explore adapting both raw and instruction-tuned LLMs using parameter-efficient fine-tuning (PEFT). We find that starting from an instruction-tuned checkpoint reduces the amount of adaption parameters needed, facilitating the adaption training process and further improving results. The combination of instruction tuning and PEFT unlocks substantial potential, achieving performance on par with full model fine-tuning on diverse text and multimodal tasks. As instruction-tuned LLMs continue to gain prevalence, non-intrusive PEFT methods like the AdaLink proposed here suffice to obtain optimized performance and emerge as a practical and effective tuning approach. Empirically, we conducted comprehensive experiments on multi-modal (captioning and VQA) tasks and natural language understanding tasks. By tuning only less than $0.02\%$ of a pre-trained language model's parameters, AdaLink reaches competitive or even better results compared to full model fine-tuning methods.

**Properties of AdaLink.** AdaLink enables efficient and scalable adaptation through its lightweight yet expressive module design. The added computational complexity grows only linearly with model embedding dimension, invariant to other model parameters. This avoids the quadratic scaling incurred by methods like prompt tuning that increase sequence length. Further, AdaLink provides flexible partial input adaptation, transforming only selected embeddings to minimize interference across modalities or tasks. The modular nature also affords configurable serving, allowing AdaLink to act as an intermediate processing unit or directly transform vocabulary embeddings. Overall, AdaLink delivers customizable and scalable task adaptation while limiting complexity overhead and preserving model architecture, making it highly promising for large-scale deployment.

## 2 BACKGROUND

**Prompt tuning.** Given a pre-trained language model with parameters $\Theta$ and a target task, full model fine-tuning can be parameter-inefficient for multiple tasks. Prompt tuning (PT) offers a more efficient and non-intrusive alternative by initializing a few learnable prompt vectors and appending them to the input embeddings without touching $\Theta$ (Lester et al., 2021a) and transformer architecture. This approach optimizes a loss function with respect to the prompt vectors and has shown to be effective. Even though prompt tuning is non-intrusive and easy to deploy, it still suffers from a big performance gap in multi-task settings (Wang et al., 2022c) and sensitivity to initialization (Lester et al., 2021a; Su et al., 2022; Zhong et al., 2022).

**Adapter and LoRA.** Alternatively, adapters (Houlsby et al., 2019) and LoRA (Hu et al., 2021) can be used to adapt LLMs for downstream tasks with a small number of additional parameters. These fine-tuning strategies introduce new parameters into LLMs in an intrusive manner. During fine-tuning, the new parameters are updated with the original LLM parameters kept frozen. Adapters and LoRA usually consist of two fully connected layers. As an example, see an illustration of adapter as shown on the right. The adapter layer uses a down projection $\mathcal{W}^{down} \in \mathcal{R}^{d \times r}$ to project input representation $x$ from model dimension $d$ to a low dimensional space $r$ (referred as the bottleneck dimension), followed by a nonlinear activation

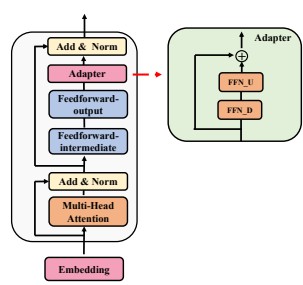

function $f(\cdot)$, and a up-projection with $\mathcal{W}^{up} \in \mathcal{R}^{r \times d}$ to project the low-dimensional features back to the original dimension.

## 3 METHODOLOGY

### 3.1 INPUT REPRESENTATIONS

**Text Representations.** For the text representation, we follow the T5 (Raffel et al., 2020) to use SentencePiece for tokenization, which breaks down the input text into subword units. Let $T = \{t_1, t_2, ..., t_n\}$ represent the input text, where $t_i$ is the $i^{th}$ token and $n$ is the length of the text. The tokenized input is passed through an embedding layer to convert into continuous vectors. Formally, this can be represented as $\mathbf{E}_{text} = \{\mathbf{e}_1, \mathbf{e}_2, ..., \mathbf{e}_n\}$, where $\mathbf{e}_i$ denotes the embedding of token $t_i$.

**Image Representations.** For the image representations, we follow PaLI (Chen et al., 2023b) to use the ViT module to produce visual embeddings. Each image is resized to a fixed size and then partitioned into non-overlapping patches with patch size $14 \times 14$. We flatten the output patch-embeddings from the ViT module as the image representations $\mathbf{E}_{image}$.

**Image-Text Representations.** Visual embeddings and text embeddings are concatenated to form the multimodal input sequence: $\mathbf{E} = \{\mathbf{E}_{image}, \mathbf{E}_{text}\}$.

### 3.2 ADALINK MODULE

In essence, AdaLink is designed around the concept of incorporating a transformation function as the link between the embedding layer and the main transformer blocks. This added layer serves as a mechanism for nuanced adaptation. The process begins with data being converted into embeddings through the embedding layer or vision modules. These embeddings are then passed through the AdaLink Modules, resulting in the transformation of the selected inputs. These transformed inputs are subsequently fed into the frozen main transformer blocks for further processing. To our surprise, we found that an adapter structure with two fully connected layers is quite effective empirically. This approach allows us to achieve competitive results without adding significant complexity, and it maintains several advantageous properties as scalable complexities and versatile deployment strategies that we will discuss in more detail in the subsequent sections.

More formally, we follow the notation from Sec. 2 to describe AdaLink, which consists of two fully connected layers. The down projection $\mathcal{W}^{down} \in \mathcal{R}^{d_{emb} \times r}$ projects input representation from the original model dimension $d_{emb}$ to a low dimensional space $r$ (referred to as the bottleneck dimension); the up-projection with $\mathcal{W}^{up} \in \mathcal{R}^{r \times d_{emb}}$ projects the low-dimensional features back to the original embedding dimension. AdaLink has the flexibility to be used as a standalone adaptation module on a per-task basis or on a per-modality basis. We introduce these two scenarios as follows and leave other potential settings for future research.

**Multi-task AdaLink.** The conventional parameter-efficient fine-tuning methods were proposed to adapt LLMs to different tasks without creating expensive copies of the original models and storage-efficient. AdaLink also enables flexibility in the granularity of task adaptation. For example, in multi-task learning scenarios, one can associate a separate AdaLink module with each task. During training, the input embeddings are selectively transformed by the task-specific AdaLink before passing through the shared transformer backbone. This targets adaptation to the nuances of each task while enabling positive knowledge transfer through the shared parameters. At inference time, the model routes the inputs through the corresponding task's AdaLink module to elicit adapted behavior for that task. The rest of the model remains unchanged, avoiding negative interference. Compared to LoRA and Adapter, AdaLink does not require to architecture modification and further reduce the engineering load extend the functions of LLMs when deploying. Compared to prompt tuning, AdaLink does not introduce additional cost to the transformer blocks with new tokens.

**Multimodal AdaLink.** In addition to per-task adaptation, AdaLink also enables flexible per-modality adaptation in multimodal settings. For models that take heterogeneous input types like text, image, audio, etc., one can associate a distinct AdaLink module with each modality. During training and inference, the embeddings for each modality get selectively transformed by their corresponding

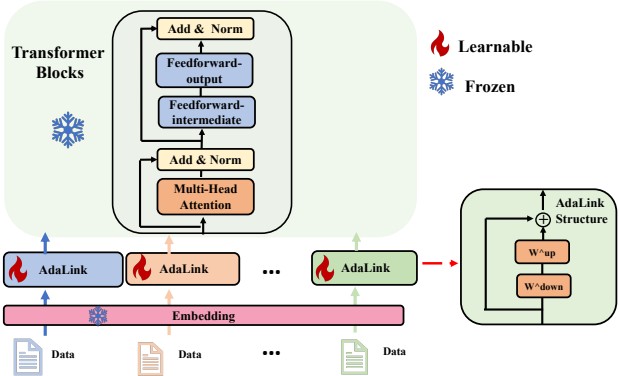

Figure 1: Overview of AdaLink. Only newly added AdaLink modules are learnable while maintaining other components frozen. The different data is first fed into embedding layer and then goes through the corresponding AdaLink respectively before main shared Transformer Blocks for adaptation per scenario.

AdaLink before fusion. A key benefit is that this modality-specific adaptation isolates interference across modalities. It also allows the modality representations to be handled independently for greater flexibility; for instance, storing them separately or fusing them at different levels. . More formally, given an input consisting of an image $\mathbf{x}^{\mathbf{image}}$ and text $\mathbf{x}^{\mathbf{text}}$, we first obtain modality-specific representations $\mathbf{E}_{image}$ and $\mathbf{E}_{text}$. These are then fed into separate AdaLink modules to get adapted embeddings

$$\tilde{\mathbf{E}}_{\text{image}} = \mathbf{E}_{\text{image}} + f(\mathbf{E}_{\text{image}} \cdot \mathbf{W}_{\text{image}}^{\text{down}}) \cdot \mathbf{W}_{\text{image}}^{\text{up}}, \tag{1}$$

$$\tilde{\mathbf{E}}_{\text{text}} = \mathbf{E}_{\text{text}} + f(\mathbf{E}_{\text{text}} \cdot \mathbf{W}_{\text{text}}^{\text{down}}) \cdot \mathbf{W}_{\text{text}}^{\text{up}}, \tag{2}$$

where $f$ indicates non-linear activation function. We find that removing non-linear activation results in only a negligible decrease in performance, thus we remove it for simplicity. The adapted modality representations $\tilde{\mathbf{E}}_{image}$ and $\tilde{\mathbf{E}}_{text}$ are concatenated to form the combined representation $\tilde{\mathbf{E}} = \{\tilde{\mathbf{E}}_{image}, \tilde{\mathbf{E}}_{text}\}$. This $\tilde{\mathbf{E}}$ is then passed into the main Transformer model for further processing. By transforming each modality separately, AdaLink provides targeted adaptation while isolating interference across modalities.

## 3.3 DISCUSSION ON PROPERTIES OF AdaLink

**Scalable Computational Costs.** Consider that we have an input with sequence length of $N$, the embedding dimension of LLMs is $d_{emb}$ and AdaLink with a rank of $r$, the added complexity is $\mathcal{O}(Nd_{emb}r)$. The computational complexity of the AdaLink remains invariant with respect to the scaling of model layers and is linearly proportional to embedding dimension of LLMs. In contrast, prompt tuning appends additional embeddings, thereby increasing the sequence length, which leads to a quadratic increase in computational complexity. This escalation in complexity can be exacerbated with the scaling of large language models (LLMs).

**Minimal Interference.** A key benefit of AdaLink is its flexibility in adapting to partial inputs, such as a subset of modalities, without requiring any changes to the main transformer architecture. The adaptation is encapsulated in the lightweight AdaLink modules that transform selected embeddings before feeding into the standard transformer blocks. Unlike methods that inject additional soft tokens, AdaLink does not modify the original input representations. This preserves the positional information of inputs like images, where spatial relationships between objects are critical. By limiting adaptation to the AdaLink modules, AdaLink allows easily adapting powerful LLMs to new scenarios.

**Configurable Serving.** AdaLink can be deployed as an intermediate processing unit as shown in Figure 1, bringing with it added complexity. Additionally, it can be utilized to transform vocabulary embeddings. In this manner, while the complexity remains constant, there is an associated increase in the storage requirements due to the addition of the embedding layer.

## 4 EXPERIMENTS

### 4.1 MULTIMODAL EXPERIMENTS

We conduct experiments on four VQA and two image captioning tasks using PaLI-X (Chen et al., 2023b), a 55B multi-modal foundational model that achieved SoTA results on a wide range of vision and language benchmarks. We demonstrate that non-intrusive PEFT achieve very competitive results compared to full model fine-tuning for a large-scale VLM like PaLI-X, especially on a multimodal instruction-tuned variant.

#### 4.1.1 BASE MODELS

**Raw checkpoint:** We refer to the PaLI-X checkpoint pre-trained per (Chen et al., 2023b) with a resolution of $756 \times 756$ as the *raw* checkpoint.

**MMIT variant:** We also experiment with a *multimodal instruction-tuned (MMIT)* variant, where we finetune the raw PaLI-X checkpoint on MMIT tasks. The MMIT tasks are created in the spirit of "Self-Instruct" (Wang et al., 2022b), taking advantage of the powerful large language models. We consider three types of tasks: (i) Long-form captioning where multiple captions are generated for each image and LLMs (Anil et al., 2023) are used to combine and summarize them into a longer and more detailed caption; (ii) Creative writing where LLMs are first used to generate novel creative writing prompts and then used to generate actual writings given the prompts based on image captions. (iii) Long-form question answering where LLMs are used to generate questions and answers with rationales given image captions. Note that these tasks collectively cover a wide variety of usecases rooted in everyday life. But they are also general in the sense that we do not expect them to be directly in-domain for the downstream tasks considered in this work. In particular, we experiment on down-stream tasks that require specific skills such as understanding scene texts and documents, or answering knowledge intensive questions.

#### 4.1.2 IMPLEMENTATION DETAILS

We compare full model fine-tuning (FT) against three types of PEFT: prompt tuning (PT) (Lester et al., 2021b), LoRA (Hu et al., 2021) and `AdaLink`. We use adafactor (Shazeer & Stern, 2018) as the optimizer. The learning rate is set to 0.03 for PEFT and 0.0001 for fine-tuning with a linear warmup and reciprocal square root decay unless otherwise specified. By default, we set the dropout rate as 0.1 to prevent over-fitting.

**Fine-tuning.** Recall that PaLI-X follows the encoder-decoder architecture where image embeddings produced by a ViT module, along with text embeddings, are fed to the multimodal encoder as one sequence. In full model fine-tuning (FT) experiments, we keep the ViT module frozen and only fine-tune the encoder-decoder backbone.

**LoRA.** We add LoRA weights on each linear layer in the multi-head attention and the MLP blocks in the encoder transformer blocks for both base models. Similar to (Yang et al., 2022), we found that adding LoRA weights in the decoder did not help the adaptation performance much at the cost of twice as many parameters. We use a LoRA rank of 16 in experiments on the raw-checkpoint and a LoRA rank of 4 in experiments on the MMIT variant.

**Prompt Tuning.** Prompt Tuning (PT) is implemented by concatenating 64 soft tunable tokens to the original input sequence, and feeding that concatenated sequence to the multimodal encoder of PaLI-X. We apply two layers of residual re-parameterization (Razdaibiedina et al., 2023) for more stable results. We use a dropout rate of 0.05 for all prompt tuning experiments as we found it to outperform the default rate of 0.1.

**AdaLink**. We insert modality-specific `AdaLink` modules to the embeddings of the text tokens and the visual tokens as a non-intrusive PEFT technique for the base model. We use a rank of 64 in all the experiments.

#### 4.1.3 IMAGE CAPTIONING RESULTS

Table 1 reports PEFT image captioning CIDEr scores (Vedantam et al., 2015) on COCO (Lin et al., 2014) and TextCaps (Sidorov et al., 2020). Within the non-intrusive PEFT family, `AdaLink`

Table 1: PEFT results on COCO captioning Karpathy test set and TextCaps captioning validation set. We report cider score for each task. `AdaLink` consistently outperforms the other non-intrusive PEFT approach (prompt tuning) and achieves competitive results to fine-tuning. [†]Recall we keep the ViT module frozen; 32B is the parameter count for the encoder-decoder backbone.

| | Non-intrusive | # params | COCO | | TextCaps | | avg. $\delta$ to FT | |
| | | | MMIT | RAW | MMIT | RAW | MMIT | raw |
|---|---|---|---|---|---|---|---|---|
| Fine-tuning (FT) | No | 32B[†] | 147 | 147.4 | 148.5 | 148.6 | 0 | 0 |
| LoRA | No | 19M | 146.8 | 146.1 | 148.6 | 147.8 | -0.05 | -1.05 |
| Prompt-tuning (PT) | Yes | 262k | 142.2 | 143.5 | 145.5 | 144.9 | -3.9 | -3.8 |
| `AdaLink` | Yes | 1.05M | 146.3 | 146.2 | 147.9 | 145.2 | -0.65 | -2.3 |

outperforms prompt tuning by about 2 cider points on average, indicating the effectiveness of directly adapting the input embeddings.

More importantly, we observe smaller gaps between `AdaLink` and FT on the MMIT variant than the raw checkpoint. This is consistent with our hypothesis that `AdaLink` can benefit more from instruction tuned base models, enabling competitive results to FT (an average of difference of 0.65). It is impressive for `AdaLink` (1.05M parameters to tune) to come within one point of full fine-tuning (32B parameters to tune). Indeed, given the much smaller number of tunable parameters, non-intrusive PEFT may suffer from less expressive power. This is perhaps less of a problem given the expressive power in large-scale base models (like PaLI-X) themselves, and partly further mitigated when base models are pre-trained on a larger variety of tasks (e.g., the MMIT variant in our experiments). Note also: while PaLI-X provides a very strong base model, with SoTA finetuning results on a wide array of benchmarks, it's not strong to the point where this level of performance can easily be achieved with zero tuning. As a reference point, on the same COCO Captions task, Chen et al. (2023b) reported a CIDEr score of 107.6 for 4-shots and 114.5 for 32-shots learning, a difference of more than 30 points to FT. Thus reaching SoTA FT performance with light-weight tuning technique like `AdaLink` is non-trivial.

While LoRA also gets better performance over the MMIT variant, the performance gap between `AdaLink` and LoRA is also smaller on this variant. Given the increasing popularity of instruction tuned LLMs, non-intrusive PEFT, especially `AdaLink`, become a strong candidate with significantly lower complexities in architecture and serving infrastructure at the cost of very minor performance degradation. As multimodal instruction tuning tasks become more comprehensive and diverse, we hypothesize there can be even smaller performance gaps between simple non-intrusive PEFT approach like `AdaLink` and intrusive PEFT or full model fine-tuning. In the case of increased base model size, complexities of non-intrusive PEFT approaches like `AdaLink` do not grow with the depth of the growing models, presenting another clear advantage in terms of practicality.

Next we present additional ablation studies on COCO Captions, again reporting results on the Karparthy test split.

Table 2: Effect of rank in `AdaLink` on the COCO captioning task.

| Rank | 4 | 16 | 64 | 256 |
|---|---|---|---|---|
| CIDEr | 144.5 | 145.3 | 146.3 | 146.3 |

**Effect of the rank.** Table 2 reports the effects of changing the ranks in `AdaLink` using the MMIT variant. We observe that the performance is not very sensitive to rank, indicating the stability of `AdaLink`. Even a rank of 4 can help the models adapt to reasonable performance, and the performance saturated at a rank of 64.

Table 3: Effect of separately adapting the input embeddings in each modality

| | Single unified `AdaLink` | | Modality-based `AdaLink` | |
| | MMIT | Raw | MMIT | Raw |
|---|---|---|---|---|
| CIDEr | 145.5 | 145.2 | 146.3 | 146.2 |

**Effect of using separate adapters for image and text modalities** Next, we compare the default modality-based `AdaLink` with separate adapters for image and text modalities to a baseline that uses one unified `AdaLink` adapter with rank 128 (twice as much as the default `AdaLink`) to adapt both visual and text tokens. Table 3 presents their performance on COCO captioning. Regardless of the base model variant used, modality-based `AdaLink` outperforms the single unified `AdaLink` by about 1 CIDEr point while using the same number of additional parameters, quantifying the benefit of modality-specific modeling, something prompt tuning struggles to achieve.

### 4.1.4 VQA RESULTS

In Table 4, we present VQA performance using PEFT on four VQA tasks: OK-VQA (Marino et al., 2019) which requires drawing upon outside knowledge, DocVQA (Mathew et al., 2021) which examines document understanding capabilities, and two scene-text understanding datasets — TextVQA (Singh et al., 2019) and ST-VQA (Biten et al., 2019). We follow standard evaluation metrics, using soft accuracy (Antol et al., 2015) for OKVQA and TextVQA and ANLS score for DocVQA and ST-VQA.

Table 4: PEFT results on four VQA tasks on the validation splits.

| | # params | OKVQA | | DocVQA | | ST-VQA | | TextVQA | | avg. $\delta$ to FT | |
|---|---|---|---|---|---|---|---|---|---|---|---|
| | | MMIT | Raw | MMIT | Raw | MMIT | Raw | MMIT | Raw | MMIT | RAW |
| FT | 32B | 66.9 | 66.1 | 82.8 | 80.0 | 79.7 | 80.2 | 70.7 | 71.9 | 0.0 | 0.0 |
| LoRA | 19M | 67.1 | 63.3 | 83.2 | 80.6 | 80.0 | 78.6 | 70.8 | 69.1 | +0.25 | -1.7 |
| PT | 262k | 66.4 | 64.9 | 82.4 | 79.7 | 79.8 | 78.3 | 70.4 | 69.7 | -0.3 | -1.4 |
| AdaLink | 1.05M | 66.8 | 63.9 | 82.9 | 78.3 | 80.0 | 77.9 | 70.2 | 67.8 | -0.05 | -2.58 |

As shown in Table 4, tuning the MMIT variant in general leads to better performance than tuning the raw checkpoint. In fact, when using the MMIT variant, the average performance differences among different tuning techniques are negligible, and `AdaLink` again emerges as an excellent choice due to its ease of serving and lower parameter counts, trailing FT by only 0.05, echoing what we saw from the captioning experiments.

It is worth noting that all three PEFT approaches, both intrusive and non-intrusive, achieved better performance on the MMIT variant, making them competitive with FT. This again points to an interesting emerging trend: the increasing power of LLMs and VLMs allows lightweight PEFT adaptation to achieve competitive performance for highly specialized use cases; moreover, this also enables non-intrusive PEFT approaches like `AdaLink` to perform competitively against intrusive ones.

### 4.2 NATURAL LANGUAGE EXPERIMENTS

**Experimental setting.** We perform experiments on a wide range of tasks including eight natural language understanding (NLU) tasks in the General Language Understanding Evaluation (GLUE) benchmark (Wang et al., 2019). We compare `AdaLink` to full model fine-tuning with various checkpoints including instruction-tuned checkpoint FLAN (Wei et al., 2021) and T5 checkpoints with additional adaption steps following (Lester et al., 2021b). Unless otherwise specified, all of the experiments in this work utilize the 11 billion parameter T5 or FLAN checkpoint as the base model.

**AdaLink implementation details.** We implement `AdaLink` in Jax for experiments. `AdaLink` uses a dimension $r$ of 4 and 256 with FLAN and T5 checkpoint in single task setting. In multi-task setting, we increase the dimensions to 256 and 1024 for FLAN and T5 checkpoints respectively. We found that most of tasks are not sensitive to rank of `AdaLink` and the performance of `AdaLink` plateaus after the modules reach a certain size. Increasing the capacity beyond this point yields diminishing returns, with little to no improvement observed in the end task metrics. The learning rate is set to 0.001 for `AdaLink`. By default, we set the dropout rate as 0.1 to prevent over-fitting.

**Single task.** The table compares full fine-tuning versus using `AdaLink` for adapting 11B T5 and FLAN checkpoints to individual GLUE tasks. For full fine-tuning, all 11 billion parameters are tuned on each task. With `AdaLink`, only the small adapter modules with 0.5-0.008 million parameters are tuned per task. We observe that `AdaLink` achieves comparable or better performance than full fine-tuning on most tasks, despite tuning far fewer parameters. For example, with the FLAN checkpoint,

Table 5: Results for NLU tasks on GLUE development set with 11B T5 and FLAN checkpoints. The best result on each task is in **bold**. Pearson refers to Pearson correlation. #Param. denotes the number of tunable adaptation parameters. FT indicates full model fine-tuning, which is usually regarded as a upper bound performance for adaptation scenarios.

| Setting | Checkpoint | Method | #Tunable Param. | MNLI Acc | QNLI Acc | SST2 Acc | QQP Acc | MRPC Acc | CoLA Mcc | RTE Acc | STS-B Pearson | Avg. |
|---|---|---|---|---|---|---|---|---|---|---|---|---|
| **Single Task** | FLAN | FT | 11B x 8 | **92.1** | 96.0 | 97.1 | 92.2 | 92.2 | 70.1 | 93.9 | 91.2 | 90.6 |
| | | AdaLink | 0.008M x 8 | 91.7 | 96.1 | 97.4 | 90.7 | 91.9 | 70.0 | **94.9** | **93.0** | **90.7** |
| | T5 | FT | 11B x 8 | 91.8 | **96.2** | 97.3 | 92.2 | 90.9 | **72.2** | 92.1 | 91.4 | 90.5 |
| | | AdaLink | 0.5M x 8 | 91.4 | 96.0 | 97.1 | **92.3** | 91.5 | 64.8 | 93.5 | 91.4 | 89.8 |
| **Multi-Task** | FLAN | FT | 11B | 91.2 | 96.1 | 97.1 | 91.9 | 90.2 | 70.2 | 93.5 | 89.5 | 90.0 |
| | | AdaLink | 0.5M | 91.8 | 95.6 | 96.8 | 90.8 | **93.1** | 64.5 | 93.1 | 92.7 | 89.8 |
| | T5 | FT | 11B | 91.7 | 96.1 | **97.5** | 90.8 | 90.0 | 65.8 | 89.9 | 87.6 | 88.7 |
| | | AdaLink | 2M | 90.1 | 93.8 | 96.0 | 91.2 | 88.0 | 60.0 | 86.3 | 89.9 | 86.9 |

AdaLink attains higher accuracy on SST-2, QQP, RTE and STS-B benchmarks. Overall, AdaLink achieves a similar average GLUE score to full fine-tuning of 90.7 using FLAN, while only tuning 0.008M adaption parameters per task. This demonstrates AdaLink's effectiveness in targeted task adaptation for large language models. The results validate AdaLink as an efficient and performant approach to adapting pretrained models to individual tasks, without compromising on model capacity. The modular architecture allows for the extension of new tasks or knowledge without the need to redevelop the main models, akin to adding patches to software during version changes.

**Multi-task.** Prior work has shown that prompt tuning approaches have optimization difficulties when applied to multiple tasks simultaneously (Wang et al., 2022c). As an input-centric method similar to prompt tuning , exploring capabilities and limits of AdaLink in the multi-task setting is informative and can help unveil the potential of this new method. AdaLink exhibits a minor gap of only 1-2% versus full fine-tuning and it achieves comparable or higher accuracy than full tuning on 6 out of 8 GLUE tasks using the FLAN checkpoint. The gap is most noticeable on the challenging CoLA task requiring complex linguistic adaptations. However, AdaLink's strong performance on most benchmarks shows that input-level tuning can effectively emulate task-specific behaviors.

Table 6: Results for NLU tasks on GLUE development set with 11B T5 and FLAN checkpoints. The performance is reported with respect to varying rank dimensions of AdaLink.

| Checkpoint | Rank $r$ | MNLI Acc | QNLI Acc | SST2 Acc | QQP Acc | MRPC Acc | CoLA Mcc | RTE Acc | STS-B Pearson | Avg. |
|---|---|---|---|---|---|---|---|---|---|---|
| **FLAN** | 2 | 91.9 | 96.1 | 97.1 | 91.0 | 91.4 | 68.7 | 94.9 | 92.8 | 90.5 |
| | 4 | 91.7 | 96.1 | 97.4 | 90.7 | 91.9 | 70.0 | 94.9 | 93.0 | 90.7 |
| | 8 | 92.0 | 96.2 | 97.3 | 90.8 | 92.2 | 68.9 | 94.6 | 93.0 | 90.6 |
| **T5** | 64 | 91.5 | 95.9 | 97.3 | 91.7 | 90.2 | 63.3 | 93.1 | 91.8 | 89.3 |
| | 256 | 91.4 | 96.0 | 97.1 | 92.3 | 90.7 | 64.8 | 93.5 | 90.6 | 89.7 |
| | 512 | 91.3 | 96.0 | 97.4 | 92.2 | 91.5 | 62.8 | 93.5 | 91.4 | 89.6 |
| | 1024 | 91.4 | 96.0 | 97.1 | 91.5 | 91.6 | 63.1 | 93.1 | 91.9 | 89.6 |

**Analysis of rank.** Our experiments demonstrate that AdaLink is not very sensitive to the rank hyperparameter. With an instruction-tuned FLAN checkpoint, a small rank of 4 achieves maximum GLUE performance, indicating compact AdaLink suffice for embedding space transformation. Increasing rank further shows negligible gains, underscoring the stability of AdaLink architecture. A larger rank is needed for the non-specialized T5 checkpoint, but performance stabilizes quickly. Overall, AdaLink attains strong adaptation with minimal parametrization across diverse initializations.

# 5 RELATED WORK

The wide scope of capabilities achieved by LLMs (Raffel et al., 2020; Brown et al., 2020; Chowdhery et al., 2022; Anil et al., 2023) and VLMs (Alayrac et al., 2022; Li et al., 2023; Wang et al., 2022a; Chen et al., 2023b) comes along with the scaling up of the parameter counts to billion level. This prohibits the conventional model deployment pipelines where different tasks own different copies of

the entire model that are served separately. We briefly introduce two means in the following sections for tackling this problem.

## 5.1 Instruction Tuning

Instruction tuning (Wei et al., 2021; Chung et al., 2022; Sanh et al., 2021; Wang et al., 2022b; Ouyang et al., 2022; Longpre et al., 2023) aims at solving a wide range of tasks using one foundation model. The entire model is fine-tuned on a large mixture of instructions formulated from the tasks of interest. (Wei et al., 2021) explore combining 62 NLP datasets with 10 instructions for each set as training data. Chung et al. (2022) further expands the scope up to 1800 tasks. The LLMs demonstrate strong capabilities in learning to interpolate the tasks used and generalize well to unseen tasks. As the instruction tuning data size is often limited, recent research proposes "Self-Instruct" (Wang et al., 2022b) that collects data by bootstrapping off their own generations, relieving the annotation burden.

**Multi-Modal Instruction Tuning**. Similar to text-only instruction tuning, Multi-Modal Instruction Tuning (MMIT) aims to jointly learn a large collection of visual language tasks. However, as most available vision-language tasks are short captioning, question-answering, and grounding for academic benchmarks that are limited in both visual scope (i.e. covered visual domains) and task scopes. Lots of tasks digress the natural use cases such as storytelling, descriptive caption generation, answering questions with explanations, etc. Therefore, most MMIT (Liu et al., 2023; Zhang et al., 2023; Dai et al., 2023; Gao et al., 2023) relies on "Self-Instruct" (Wang et al., 2022b) protocols that create training tasks automatically.

## 5.2 Parameter Efficient Fine Tuning

Instead of deploying specialized full models, recent research investigates more on the parameter-efficient fine-tuning (PEFT) that only adapts a tiny portion of parameters, keeping most of the parameters frozen. We categorized the PEFT approaches into intrusive and non-intrusive approaches.

**Intrusive PEFT** makes direct changes to the model architectures, usually to the transformer blocks. Layer-wise prompt tuning (Liu et al., 2021) and LLaMA (Zhang et al., 2023) prepend tunable tokens to the transformer blocks' inputs. Adapters (Houlsby et al., 2019; Pfeiffer et al., 2020, 2021) insert low-rank MLPs in each block. LoRA (Hu et al., 2021) takes a step further and adds low-rank weights in each linear layer within the self-attention and the MLPs. Though the intrusive PEFT approaches offer more flexibility in design, they introduce significant challenges in model deployment where the adaptation weights need to be transferred to the internal architecture. Besides, the size of the tunable parameters still grows proportionally to the model size.

**Non-intrusive PEFT** is input-centric which keeps the core transformer blocks frozen, including both the pre-trained parameters and the computation graph. Prompt tuning is the classic example where the tunable tokens are prepended to the word embeddings before being fed into the transformer blocks. However, Experiments show that prompt tuning struggles with optimization difficulties Razdaibiedina et al. (2023), requiring a large number of training examples. We propose `AdaLink` that adapts the input embeddings using low-rank MLPs, taking the benefit of "zero init" that avoids the disturbance at the beginning of training. We show that the `AdaLink` achieves competitive results as the full-model fine-tuning with scaling up of the model size.

## 6 Conclusions

In this paper, we examine the influence of scaling up both model parameter counts and pre-training tasks to parameter efficient tuning (PEFT) on both text only and multimodal down-stream tasks. We show that the performance gap between full model fine tuning and PEFT are significantly narrowed with the help of both. This indicates the increasingly powerful LLMs and VLMs only require a slight adaptation, and input-centric non-intrusive PEFT is often enough to obtain optimized performance and enjoys the ease of deployment and constant size with respect to model depth. We also introduce `AdaLink` that achieves better adaptation performance than prompt tuning within the non-intrusive PEFT family.

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
