# OpenReview forum: "Non-Intrusive Adaptation: Input-Centric Parameter-efficient Fine-Tuning for Versatile Multimodal Modeling"
_ICLR.cc/2024/Conference — ICLR 2024 Conference Withdrawn Submission_

### Official Review · Reviewer_cRLW · 2023-11-01

**Soundness:** 3 good
**Presentation:** 3 good
**Contribution:** 2 fair
**Rating:** 3
**Confidence:** 4

**Summary:**

This paper focuses on Parameter-efficient Fine-tuning (PEFT) techniques for adapting and deploying large language models (LLMs) and vision language models (VLMs). The paper introduces AdaLink, a non-intrusive PEFT technique, and demonstrates that it achieves competitive performance compared to state-of-the-art intrusive PEFT methods like LoRA, as well as full model fine-tuning, across various text-only and multimodal tasks.

**Strengths:**

1. **Parameter Efficiency**: AdaLink offers a non-intrusive method for fine-tuning large models, which is beneficial for parameter efficiency. This is crucial when dealing with models of such a large scale, as it enables easier adaptation and deployment without the need to modify and retrain the entire model.

2. **Competitive Performance**: The paper demonstrates that AdaLink is capable of achieving competitive performance compared to both state-of-the-art intrusive PEFT methods and full model fine-tuning across a variety of tasks. This highlights the effectiveness of AdaLink as a viable option for model adaptation.

**Weaknesses:**

1. **Average Performance Compared to Baselines**: The submission demonstrates that the performance of the proposed method is mediocre when compared to other benchmarked techniques. This raises questions about the significance of the method.

2. **Lack of Distinction from Existing Prompt Tuning Methods**: The simplicity of the proposed method, while potentially beneficial for ease of implementation, also means that it does not significantly differentiate itself from existing prompt tuning techniques, such as CoOP.

3. **Absence of Motivation and Inspiration**: The submission fails to provide a clear motivation or inspirational insight behind the proposed method, leaving readers with unanswered questions about the rationale and the innovative aspects of the approach. This lack of a compelling narrative or a strong theoretical foundation could potentially limit the impact and the appeal of the work to a broader audience.

**Questions:**

Please refer to the weakness

---

### Official Review · Reviewer_ZwpG · 2023-11-02

**Soundness:** 3 good
**Presentation:** 3 good
**Contribution:** 2 fair
**Rating:** 3
**Confidence:** 4

**Summary:**

The paper proposes AdaLink, an parameter efficient adaptation module for large multi-modal foundation models. AdaLinks comprise of down- and up- projections and identity connections, forming a bottleneck-shaped residual architecture. AdaLinks are inserted exclusively to the middle between token embeddings and the transformer, and are modal-aware, i.e., a separate set of weights is used for each supported input modality. The claimed advantages of such non-intrusive adapters include low computational costs, minimal interference between tasks, and less engineering efforts to serve the model. The adapted models are evaluated on various tasks, including captioning and VQA for multi-modal understanding, and NLU tasks from GLUE for natural language understanding, and obtained results comparable to full-finetuning and other parameter-efficient fine-tuning baselines.

**Strengths:**

* The writings and illustrations are clear and easy to follow. The experiments cover a wide range of tasks.

* Among non-intrusive adaptation methods, the proposed method can outperform the baseline prompt tuning.

**Weaknesses:**

* **The advantages of being non-intrusive is not persuasive.** This is by far the most concerning point of mine. Since the performance of AdaLinks still fall behind intrusive methods (e.g., LoRA) in many experiments, it is very important that AdaLinks clearly excel in some other aspects. I list my questions for each of the point 1 and 3 mentioned in section 3.3 as follows:
  * *Scalable computational cost.* (1) The statement *prompt tuning ... increasing the sequence length, which leads to a quadratic increase in computational complexity* is misleading. The *added complexity* of prompt tuning is actually $O(Ld_{emb}^2r + LNd_{emb}r)$, with the $r$ being number of prompts, $L$ being transformer depth, first item being *added complexity* in linear layers and the second term being *added complexity* in attention layers, from which we can see that the *added complexity* is in fact *linear* to the sequence length if measured by the same standard (i.e., by consider only the additional computation on top of the backbone computation) as used in the previous statement about the paper's own method. (2) Nevertheless, the total cost of the adapted network will be dominated by the backbone cost for most (if not all) known adaptation methods, in which case further optimizing the additional cost of adaptation will bring very limited advantages according to Amdahl's law. (3) It is indeed possible that for some methods, the theoretical additional cost is small but the actual additional cost on hardware is large, in which case the paper needs to give some fundamental reasons why this is difficult to resolve (e.g., due to hardware limitations), ideally with measured performance numbers (e.g., throughput or latency on actual hardware).

  * *Configurable Serving.* The ease of deployment need to be compared to the other well-known adaptation methods in more details. For example: (1) LoRA weights can be merged into the backbone weights as $W_{adapt} = W_{backbone} + W_{up} W_{down}$ without any architectural change. On what specific infrastructure will AdaLink be easier to deploy than merged LoRA with exactly the same architecture before and after PEFT? (2) Adapters [1] are usually inserted before / after / parallel to a whole transformer block. What are the specific reasons making them much harder to deploy than AdaLinks [e.g., What are the specific cases in which *adaptation weights (are difficult) to be transferred to the internal architecture* (quoted from section 5.2)? Or why a fused operator needs to cross the block boundary to make intrusive adapters difficult to insert?]

* **Limited novelty.** Other than introducing the modal-specific adaptation for different tokens, the concept and architecture are very similar to the original adapters [1]. Applying modal-specific projections is also among the first attempts of adapting language models to multiple modalities (e.g., [2, 3] but far from being complete).

* **Community accessibility.** Most results in the paper reported using PaLI-X, which I believe is still not open-sourced at the time of this review. Thus, the community may face difficulties reproducing the results or further developing the method. It would be helpful if the paper could also include some open-source model results for future reference.

[1] Houlsby, Neil, et al., Parameter-efficient transfer learning for NLP., ICML 2019.

[2] Eichenberg, Constantin, et al., MAGMA - Multimodal Augmentation of Generative Models through Adapter-based Finetuning, EMNLP 2022.

[3] Yang, Antoine, et al., Zero-shot video question answering via frozen bidirectional language models, NeurIPS 2022.

**Questions:**

My questions currently focus on section 3.3 or weakness 1. To summarize, I'm most interested in the details about a specific and broadly applicable case (hardware or infrastructure architecture) where AdaLinks are the only favorable / feasible way of adaptation compared to other methods. I would raise my rating once the explanation about this is clearer.

---

### Official Review · Reviewer_bjPo · 2023-11-06

**Soundness:** 3 good
**Presentation:** 3 good
**Contribution:** 2 fair
**Rating:** 5
**Confidence:** 3

**Summary:**

This article proposes an adaptation strategy for large language models and large multimodal models.
It consists of a two-layer MLP that processes the transformer’s input tokens, after their modality-specific encoders.  The authors evaluate their method on various vision-and-language tasks (captioning, VQA), and textual tasks, using the PALI-X model for multimodal tasks, and FLAN or T5 for NLP tasks. They show that this method reaches competitive performances compared to full fine-tuning or more complex PEFT strategies like LORA.

**Strengths:**

The paper is clearly written and easy to follow.

They evaluate their method on many vision-and-language tasks as well as textual tasks.

The method is simpler and has fewer parameters than the alternative stat-of-the-art method LORA but reaches similar performances.

**Weaknesses:**

## intrusive vs non-intrusive

I don't think the distinction between intrusive versus non-intrusive methods is obvious, and it doesn't seem a good argument for the method.
In practice, all methods have a set of weights that they fine-tune, and I don't see how their position in the network makes a huge difference in terms of deployment, as all the weights will be packaged in a single model. It might be slightly easier to adapt to different network architectures, but the AdaLink makes the assumption that the input is a set of tokens, and is only evaluated on transformer architectures, so this is hard to assess. Maybe the authors should evaluate it on different kinds of architectures to make this point stronger.

Therefore, I think this distinction should not be highlighted in the paper.
It would be more interesting if the comparison would focus on quantitative metrics, like additional FLOP to the forward/backward passes or the number of parameters (this is reported already), and the experimental results, which are convincing.

## Novelty and related works
Some papers have proposed similar approaches to adapt textual LLM and visual models together into multimodal models, which are not cited in this paper:
- BLIP-2 [1], which uses a light transformer model to bridge between the visual and the textual models
- Some papers evaluate the use of a *single* linear layer [2,3].

Those methods have similarities to the proposed work, therefore the novelty of this work is not as important as claimed. The differences with those works are: (a) they use an already trained multimodal model instead of two frozen unimodal models (b) they use a two-layers MLP instead of a transformer or a single-layer MLP.
Those works should be cited in the related work, and if possible evaluated on the same models.

## Nonreproducible multimodal experiments
The multimodal model is not available to the public. It would be nice to have the same experiments on an open multimodal model like OpenFlamingo or Idefics.

[1] BLIP-2 https://arxiv.org/abs/2301.12597

[2] Limber https://arxiv.org/abs/2209.15162

[3] eP-ALM https://arxiv.org/abs/2303.11403

**Questions:**

1 - How did you choose the rank of LORA methods? It is much lower than the rank of AdaLink.

2 - It is hard to understand whether the difference between LORA and AdaLink comes from the position of the layer, or from the number of parameters. Maybe a single LORA layer, with a similar number of weights, would have similar scores.
Could you add a bigger ablation on the size of the two-layer network, or on the number of LORA parameters?

3 - Trying your method to adapt two frozen unimodal models (vision and text) would be interesting.

4 - Focus on quantitative metrics between the two methods to explain why AdaLink is better than LORA, rather than the unclear "intrusive" vs "non-intrusive".

5 - Evaluate with other, open-source multimodal models

I am willing to increase my score after the rebuttal

---

### Official Review · Reviewer_R26q · 2023-11-09

**Soundness:** 3 good
**Presentation:** 3 good
**Contribution:** 1 poor
**Rating:** 3
**Confidence:** 4

**Summary:**

The authors present a method to optimize an encoder-decoder generative model across multiple new tasks with only a fraction of the parameters it would take to finetune the model. Their model called “AdaLink” is made of a small MLP that is plugged between the image encoder and the large language model of a PALI-X model or before the decoder of a T5/FLAN model to stir the outputs of the full backbone toward the respective new tasks.
They evaluate their results on multiple VQA and captioning datasets with a raw PALI-X model and an instruction tuned one, and on NLU for the FLAN and T5 models.
They compare their method against three baselines: full finetuning, LORA and prompt tuning and show that their method perform better and that instruction tuned models get even better results than non-tuned ones.

**Strengths:**

- Clear presentation
- Ablations are good
- The conclusion that instruction tuned model are easier to tune is interesting

**Weaknesses:**

- Limited novelty: a lot of papers already use this kind of architecture for image-text generative models.

     1) Visual Instruction Tuning, Liu et al. -> They use a linear projection between the image encoder and the large language model. They freeze the LLM in a first step, showing good results as an adaptation layer, then fine tune the LLM in a second step.
     2) eP-ALM: Efficient Perceptual Augmentation of Language Models, Shukor et al. -> They use a linear projection between the image encoder and the frozen LLM but don’t use multiple image tokens. Instead they concatenate the projected image token to each layer of the LLM. They show that this method works well across modalities too.
     3) MiniGPT-4: Enhancing Vision-Language Understanding with Advanced Large Language Models, Zhu et al -> They use a linear projection between the image encoder and the frozen large language model.

- It is hard to tell the gain those PEFT methods allow to make starting from the original model. I think that the performances of the original/MMIT Pali-X in 4 and 32 shots should be a baseline on table 1 and table 4 and the same 4/32 shots baseline for FLAN/T5 should be in table 5.

**Questions:**

- While the gain on COCO, going from ~110 in 32 shots to ~140 with PEFT seems impressive, it is hard to tell if the model was already really good (especially the MMIT one) and is just overfitting on COCO. Adding the 4/32 shots baseline would help. You could also do multi-task analysis that is performed on the text evaluations (table 5) between COCO and TextCaps in table 1 and between all VQA datasets in table 4 would help to disentangle overfitting to the specific vocabulary/text length/format of the task from real understanding of the task.

- AdaLink is really good at tuning the model to one task, and is pretty good for multiple NLU tasks (table 5), but it might lack expressivity against other methods such as LoRA in the multi-tasks setup. While prompt tuning is hard in the multi-task case, could you add the LoRA baseline to the Table 5 ? The rank ablation from table 6 applied to the multi-task of Table 5 could also help us understand the effect of the number of tasks on the best rank.